



# Spatially heterogeneous effect of the climate warming on the Arctic land ice

Damien Maure[1,2], Christoph Kittel[2,1], Clara Lambin[1], Alison Delhasse[1], and Xavier Fettweis[1]

[1]SPHERES research unit, Geography, University of Liège, Liège, Belgium
[2]Université Grenoble Alpes, CNRS, IRD, Grenoble INP, IGE, 38000 Grenoble, France

**Correspondence:** damien.maure@uliege.be

**Abstract.** Global warming has already substantially altered the Arctic cryosphere. Due to the Arctic warming amplification, the temperature is increasing more strongly leading to pervasive changes in this area. Recent years were notably marked by melt records over the Greenland Ice Sheet while other regions such as Svalbard seem to remain less influenced. This raises the question of the current state of the Greenland Ice Sheet and the various ice caps in the Arctic for which few studies are available. We here run the Regional Climate Model (RCM) Modèle Atmosphérique Régional (MAR) at a resolution of 6 km over 4 different domains covering all Arctic land ice to produce a unified Surface Mass Balance product from 1950 to present day. We also compare our results to large-scale indices to better understand the heterogeneity of the evolutions across the Arctic and their links to recent climate change. We find a sharp decrease of Surface Mass Balance (SMB) over the western Arctic (Canada and Greenland), in relationship with the atmospheric blocking situations that have become more frequent in summer, resulting in a 41% increase of the melt rate since 1950. This increase is not seen over the Russian Arctic and Svalbard permanent ice areas, where melt rates have increased by only 9% on average, illustrating a heterogeneity in the Arctic SMB response to global warming.

## 1 Introduction

The warming amplification of the Arctic has led to a temperature rise of +3.8 degrees on average poleward of 66.5°N since 1979, 4 times larger than the global average (Rantanen et al., 2022). While this warming contributes to a higher melting rate of glaciers and ice caps (e.g., Fettweis et al., 2017; Noël et al., 2018), it has also raised the atmospheric humidity leading to more solid precipitation in winter (Przybylak, 2002; Førland et al., 2002). In combination with large-scale atmospheric circulation variations, the average melting and precipitation rates could modify the surface mass balance (SMB) of the Arctic land ice (i.e, the Greenland Ice Sheet, Arctic ice caps and major perificial glaciers).

The SMB, which is the difference between the total amount of precipitation (solid and liquid) and the ablation by meltwater runoff and evaporation/sublimation, is a component of the total ice mass budget of permanent ice areas together with the ice discharge driven by the ice dynamics (Note that the definition we use as SMB is formerly defined as Climatic Mass Balance in Cogley et al. (2010)). However, the Arctic SMB is more sensitive to quick climate variations and its importance in the total mass budget is expected to increase relative to the ice discharge, at least over the Greenland Ice Sheet (Fürst et al., 2015).



Furthermore, the combined Arctic permanent ice areas (excluding the Greenland Ice Sheet) are the major contributor to sea level rise after the ice sheets (Gardner et al., 2013; Moon et al., 2020).

While the warming trend is global, the different studies carried over the Arctic indicate a regional heterogeneity in the response of SMB to the climate of the last decade. The higher frequency of blocking anticyclonic events has increased the summer melt rate over the Greenland Ice Sheet or Canadian ice caps (Fettweis et al., 2013; Lenaerts et al., 2013; Noël et al.,

2018; Fettweis et al., 2017; Topál et al., 2022). On the contrary, recent North Atlantic cooling has decreased glacier mass loss rates in Iceland (Noël et al., 2022). In Svalbard, atmospheric circulation changes has prevented significant changes in SMB (Lang et al., 2015; Van Pelt et al., 2019).

High-resolution dynamical downscaling has enhanced the estimations of SMB across the Arctic by providing continuous results in space and time compared to in situ observations (and satellite data). However, there still lacks a unified estimate over

all the permanent land ice areas of the Arctic using the same method. Moreover, SMB estimates over the Russian High Arctic remain very scarce. Here, we present the results from a series of dynamical downscaling simulations, at high resolution (6km), covering all the Arctic regions with permanent Arctic land ice (Baffin, Devon, Ellesmere, Iceland, Svalbard, Greenland, Franz Joseph Land, Nova Zembla and the Russian High Arctic Islands), using the Modèle Atmosphérique Régional (MAR) Regional Climate Model (RCM). The aim of the study is to 1) present a unified SMB product derived through the same method over all

the Arctic, and to 2) highlight the links between SMB changes over different regions and general climate patterns.

## 2 METHODS

### 2.1 MAR

MAR is a 3D atmosphere-snowpack RCM initially designed for polar regions (Gallée and Schayes, 1994). It has been used in multiple studies and proven to be reliable to reconstruct the recent SMB changes over the Greenland (Fettweis et al., 2017,

2020) and Antarctic (Agosta et al., 2019) ice sheets or smaller ice caps (Svalbard, Lang et al., 2015).

MAR resolves the primitive equations using the hydrostatic approximation and has a vertical sigma coordinate system. MAR also includes the 1D surface scheme SISVAT (Soil Ice Snow Vegetation Atmosphere Transfer; De Ridder and Gallée, 1998; Ridder and Schayes, 1997; Gallée and Duynkerke, 1997; Gallée et al., 2001; Lefebre et al., 2003) which describes the surface properties and their evolution through their interactions with the atmosphere. The snow/ice module of SISVAT describes the

snowpack metamorphism and properties (such as temperature, liquid water content and density) of the 20 first meters of permanent ice areas divided into 30 layers of snow, firn or ice. Since MAR is here not coupled with an ice sheet model, the topography and ice extent are fixed in the model throughout the entire simulations. Pixels are considered as ice-covered only if they have at least 50% of their area covered by ice.

In this study, MAR version 3.11.5 (hereafter MARv3.11.5) is used to reconstruct SMB changes over the Arctic ice caps

and ice sheet. The improvements of this version are described in Kittel et al. (2021). A general summary of the modules and schemes used in MAR can also be found in Fettweis et al. (2017). Fig.1 presents the 4 integration domains (without the relaxation zone) used to run MAR over all the permanent Arctic ice areas, at a 6-km horizontal resolution using 24 vertical

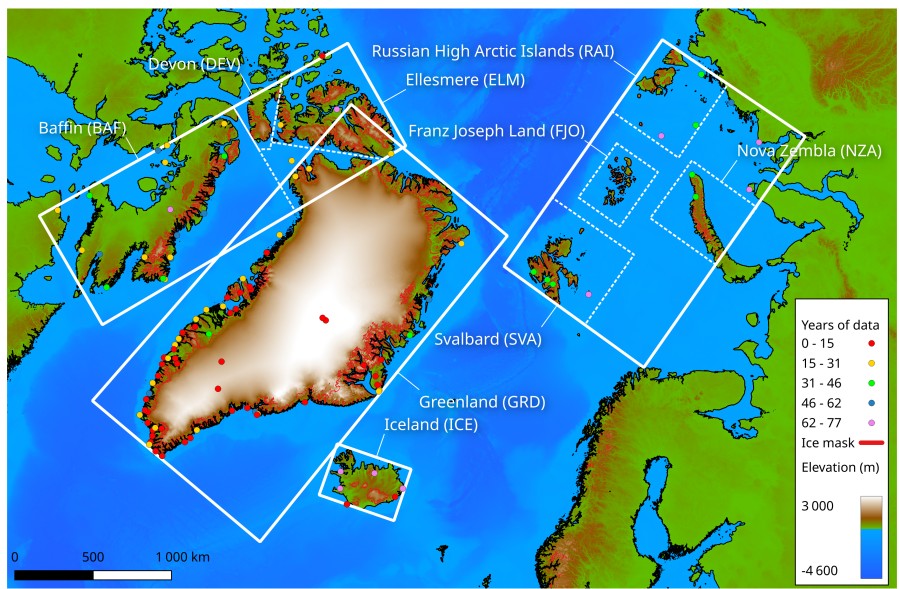

**Figure 1.** MAR domains used over the Arctic (white solid boxes) and integration sub-regions for analysis (dashed boxes). AWS locations are shown with a dot colored as a function of the years of data they provide.

layers in atmosphere with a first level at 2 meters above surface. We used 4 different integration domains in order to reduce the computational cost of a 70-years-long simulation, enabling for such a high horizontal resolution. The model parameters and
set-up are kept the same over all the domains. While a 6km resolution might be too low to fully resolve the elevation of the smaller ice bodies, the average hypsometry of the model grid ice pixels remains close to observations on every sub region (see Fig.S1). The biggest discrepancy can be seen in Franz Joseph sub region where our grid is on average underestimating the real ice elevation, though it remains a relatively small bias and most likely does not have a major influence on the results.

## 2.2  Reanalysis

The ERA-5 reanalysis (Hersbach et al., 2020) is used as forcing fields to prescribe MAR boundary conditions every 6 hours at each vertical level and over ocean (temperature, u- and v-components of the wind, humidity, surface pressure, and at the surface sea ice concentration and sea temperature). We chose ERA-5 because it has the advantage to be continuous from 1950 to present day, and is performing well over Greenland (Delhasse et al., 2020).

## 2.3  Data & evaluation methods

MAR has been often used over Greenland (e.g., Fettweis et al., 2020; Lambin et al., 2022) and Svalbard (Lang et al., 2015) but less frequently over Canada, Iceland or the Russian Arctic. As our study deals with 3 new MAR domains, more attention is given to the evaluation of the results against field observations. First, the simulations are evaluated for their performance in





reproducing real atmospheric conditions (in particular the 2m temperature, pressure and wind speed). Then, the reconstructed SMB is compared to the few observations available: satellite altimetry (from Hugonnet et al., 2021) to compare the regionally-integrated SMB from 2000 to 2020 over land-terminating glaciers, along with the SMB dataset from Machguth et al. (2016) over the Greenland Ice Sheet, available on the PROMICE website.

### 2.3.1 Evaluation of the atmosphere

Over the different domains, 102 automatic weather stations (AWSs) were used to evaluate the MAR simulations. The localisation of the AWSs is shown in Figure 1. Daily average values were used to compare observations to MAR simulations. For the modeled values, daily means were extracted as a distance-weighted mean between the 4 nearest MAR pixels. To avoid bias coming from ocean pixels (where the SST is prescribed into MAR from ERA5), only land MAR pixels were considered for the evaluation. Finally, the AWSs with an elevation difference of more than 200m with the 4-nearest-pixels average were excluded to avoid artificial biases driven by the elevation difference (16 excluded in total). Mean bias, root mean squared error (RMSE), centered root mean squared error (CRMSE), and correlation (r) between observed and modeled values were computed.

### 2.3.2 SMB evaluation

As precipitation and snow surface processes are the most challenging variables to represent in climate models, large biases can arise between models and observations when simulating the SMB. It is crucial to evaluate the modeled SMB over the different regions, although direct observations remain scarce, especially over the Russian Arctic.

Hugonnet et al. (2021) developed a global product of glacier elevation change from 2000 to 2019, using NASA's Advanced Spaceborne Thermal Emission and Reflection Radiometer (ASTER). Their glacier mass change product consists of monthly mass loss estimates integrated over sub-regions of the Randolph Glacier Inventory (RGI). It contains Mass Balance (MB) estimates for all the land ice in Canada, Iceland, Svalbard, Russian Archipelagoes and the Greenland periphery. Because the MB is the difference between the SMB and the dynamical iceberg discharge (in the case of marine terminating glaciers), we selected data for only land-terminating glaciers, using their classification in the RGI 6.0. Annual modeled SMB was then integrated over all the glaciers to be compared with MB estimates. This altimetry dataset is useful in order to evaluate the SMB over large remote regions of our study, where very sparse in situ observations are available. We use the in situ SMB dataset from Machguth et al. (2016) to evaluate MAR as done in Fettweis et al. (2020) over Greenland, as the mass loss by iceberg discharge over the Greenland Ice Sheet is significant compared to smaller Arctic ice caps. This dataset contains historical SMB measurements from more than 3000 stakes over the ice sheet. It is quite different from the evaluation using the Hugonnet et al. (2021) MB product (annual spatially integrated data), so the results will not be intercomparable, but gives another estimate on the performance of MAR.





| | | Annual | | | | Summer | | | | Winter | | | |
|---|---|---|---|---|---|---|---|---|---|---|---|---|---|
| | | Mean obs | Bias | CRMSE | r | Mean obs | Bias | CRMSE | r | Mean obs | Bias | CRMSE | r |
| T2m [°C] | Canada | -10.3±11.0 | -0.7 | 2.7 | 0.97 | 3.0±2.8 | 0 | 2 | 0.77 | -21.0±6.2 | -0.7 | 2.8 | 0.9 |
| | Iceland | 4.3±4.6 | -1.3 | 1.3 | 0.96 | 9.1±2.0 | -0.7 | 1.3 | 0.82 | 0.4±3.7 | -1.6 | 1.4 | 0.94 |
| | Greenland | -4.6±8.2 | -1.3 | 2.7 | 0.95 | 4.5±2.8 | -0.3 | 1.9 | 0.79 | -12.7±6.0 | -2 | 3 | 0.88 |
| | Svalbard | -5.1±9.1 | -2.6 | 2.4 | 0.97 | 4.6±2.6 | -3.1 | 1.3 | 0.87 | -12.4±8.1 | -2.1 | 3 | 0.93 |
| | Russia | -11.7±12.2 | -0.7 | 3 | 0.97 | 1.7±3.3 | -0.9 | 1.7 | 0.85 | -23.3±8.5 | -0.6 | 3.7 | 0.9 |
| P2m [hPa] | Canada | 1011.6±11.2 | -17.3 | 2.1 | 0.98 | 1010.9±7.7 | -16.3 | 1.5 | 0.98 | 1009.1±13.3 | -17.9 | 2.3 | 0.99 |
| | Iceland | 1006.3±13.8 | -7 | 1.1 | 0.99 | 1010.6±8.4 | -7 | 0.7 | 0.99 | 1000.8±16.7 | -7 | 1.2 | 0.99 |
| | Greenland | 1012.5±11.8 | -36.7 | 3.4 | 0.93 | 1013.8±7.8 | -38.1 | 2.4 | 0.93 | 1009.0±14.0 | -39 | 3.7 | 0.94 |
| | Svalbard | 1007.6±11.6 | -34.8 | 1.4 | 0.99 | 1010.2±7.7 | -33.7 | 0.8 | 1 | 1003.4±13.7 | -35.4 | 1.6 | 0.99 |
| | Russia | 1011.4±11.9 | -8.1 | 1.8 | 0.99 | 1011.2±8.5 | -7.8 | 1.4 | 0.99 | 1011.0±14.4 | -8.1 | 2 | 0.99 |
| WS [$m\,s^{-1}$] | Canada | 3.7±2.6 | 0.3 | 2.3 | 0.66 | 3.2±2.4 | 0.1 | 2 | 0.74 | 4.0±2.7 | 0.4 | 2.4 | 0.61 |
| | Iceland | 5.1±3.0 | -0.4 | 2.2 | 0.75 | 4.1±2.3 | -0.4 | 1.8 | 0.72 | 5.9±3.4 | -0.3 | 2.5 | 0.72 |
| | Greenland | 4.4±2.9 | -0.4 | 2.3 | 0.64 | 3.5±2.0 | -0.5 | 1.8 | 0.59 | 5.1±3.3 | -0.3 | 2.6 | 0.65 |
| | Svalbard | 4.5±2.9 | 0.3 | 2.2 | 0.71 | 3.9±2.1 | -0.2 | 1.9 | 0.6 | 5.3±3.4 | 0.5 | 2.4 | 0.73 |
| | Russia | 6.3±3.7 | -1.5 | 2.5 | 0.75 | 5.5±2.8 | -1.1 | 2 | 0.71 | 7.0±4.3 | -1.6 | 2.7 | 0.78 |

**Table 1.** Evaluation results (Bias, Centered Root Mean Squared Error (CRMSE) and Correlation coefficient (r)) over all regions, annually and seasonally. Standard deviation is provided as a +/- value. (19 AWSs in Canada, 6 in Iceland, 69 in Greenland , 4 in Svalbard and 7 in Russian Arctic.)

## 3 Evaluation

### 3.1 Climate evaluation

Table 1 presents the resulting mean bias, CRMSE, and r over all regions for the near-surface temperature, pressure, and wind
speed. The results are computed annually and seasonally (JJA for summer and DJF for winter) and for each MAR domain. As the dataset for Svalbard and Russian Archipelagoes are inside the same domain, we separated them for the evaluation as the observational datasets are different. The height measurement for wind speed was not always available. We then use the 2m wind speed from MAR for the comparison, potentially leading to inherent biases.

The correlation coefficient between MAR and observed 2-meters pressure (P2m) is mostly larger than 0.9 over all regions.
The high negative bias over Svalbard and Greenland is imputable to some of AWSs observations being corrected to sea level pressure, whilst model values are computed at the pixel elevation. This difference does not influence the correlation which is the only relevant statistical value concerning pressure. The 2m temperature is also reproduced very well in each domain (the annual correlation coefficient is artificially driven up because of the seasonal cycle). There is however a general negative bias compared to observations. Moreover, the temperature is better reproduced in winter than in summer (r = 0.91 vs 0.81),
because temperature variability is lower as the near-surface temperature is close of 0°C most of the time and less driven by the general circulation dynamics than in winter. Finally, the modeled 2m wind speed has a bias lower than $\pm 1.5\,m\,s^{-1}$ and the performances of the MAR reconstruction are homogeneous over all domains. However, wind speed observations are

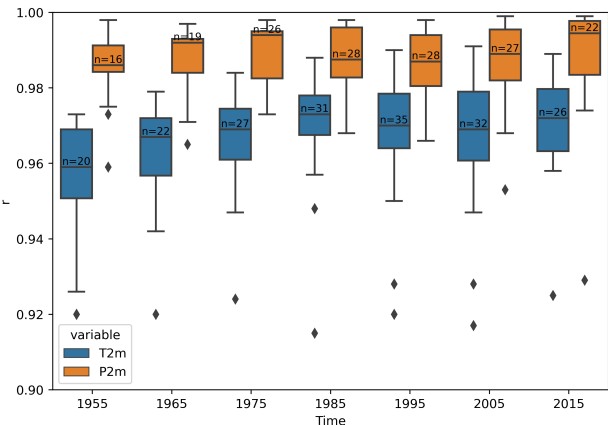

**Figure 2.** Time evolution of the near-surface Pressure (P2m) and Temperature (T2m) correlation coefficient between observation and simulation at the annual time scale. Boxes show median & quartiles of the correlation distribution amongst stations, whiskers extent to show the rest of the distribution outside of outliers (diamonds).

particularly sensitive to local site effects, which are not resolved at a resolution of 6km (as seen by the correlation values lower than 0.7, Lambin et al. (2022)). Moreover, we do not have the information of the height of the measurement which can also

influence the comparison. The quality of the reanalysis products (ERA-5) depends largely on the number of observations that were assimilated. Because our study goes up to 1950, it is worth evaluating the precision as a function of time: the further we go back, the fewer observations available. For example, ERA-5 prior to 1979 has been shown to be performing poorly above the Antarctic, because of the scarcity of satellite observations over the continent (Marshall et al., 2022).

While we could expect a better agreement after 1979 in our ERA5 forced MAR simulation due to the assimilation of

satellite data in the reanalysis, there is not a significant evolution of the correlation coefficient as a function of time for the P2m (Fig.2). The latter is constant at approximately r = 0.99 from 1950 to the present day. The analysis nevertheless reveals a slight increase of the correlation coefficient in the 2m temperature, from 0.96 in 1955 to 0.97 in 1985 as a results of satellite datasets assimilation (in particular sea ice cover (SIC) and sea surface temperature (SST)) after 1979: this strongly influences the reconstructions elsewhere, mostly where observational data was scarce (Marshall et al., 2022). However, the good amount

of older observations in the Arctic (compared to the Southern Hemisphere) explains the good performance of MAR forced by ERA-5 before 1979 (Hersbach et al., 2020); and even before 1957 the International Geophysical Year (see Table S1).

## 3.2 SMB

Figure 3 shows the statistical distribution of annual modeled SMB values (mod) and MB satellite observation (obs) estimates over land-terminating glaciers in all sub-regions, for the period 2000–2020. There are some biases in the annual mean val-

ues, positive over Svalbard, Greenland periphery and Ellesmere Island, negative over Baffin, Devon and Iceland. The main difference is in the variability of the values, where modeled interannual variability is systematically higher than the observed





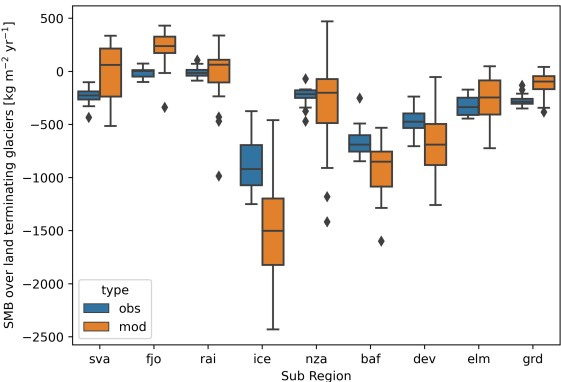

**Figure 3.** Statistical distribution of average yearly SMB values over land terminating glaciers according to the RGI 6.0, between 2000 and 2020, for modeled values (in orange) and observed (MB estimates from Hugonnet et al. 2021, in blue). (Note that Greenland (grd) only includes peripheral glaciers). Boxes show median & quartiles of the distribution amongst years, whiskers extent to show the rest of the distribution outside of outliers (diamonds).

one. This could be related to the lower interannual variability of altimetry products because of snowpack densification and ice dynamics (Li et al., 2023).

Land-terminating glaciers represent only a small fraction (10% when accounting for the Greenland Ice Sheet, 43% without) of all the ice areas studied here. The main bias of this evaluation comes with the integration of the 6-km MAR pixels over small glaciers (especially with small ice tongues) with a strong spatial SMB gradient, or very sensitive to site effects. Finally, while the RGI is generally precise in the classification of land/marine-terminating glaciers, it is sometimes less accurate (as in northern Svalbard for example), which could explain a slight positive bias of the simulated SMB values as some ice discharge would be included in the observation dataset.

Over the Greenland Ice Sheet, the evaluation using the PROMICE dataset yields a correlation of 0.93 between the model and the observations. The average bias is +0.03 m w.e. yr-1 (for an average observational value of -0.86 m.w.e yr-1) and the RMSE is 0.43 m w.e yr-1.

## 4 Results

Our simulations show that the Arctic experiences an overall yearly SMB anomaly of –96.4 Gt yr-1 over 2000—2020 compared to the reference period of 1950–1970. This value becomes even more negative when considering the recent past evolution, with an anomaly of -154 Gt yr-1 between (1975–1995) and (2000–2020). While this decrease is mainly driven by Greenland, as its runoff has increased by 35% between (1975–1995) and (2000–2020), it has on average increased by 45% on the other regions.

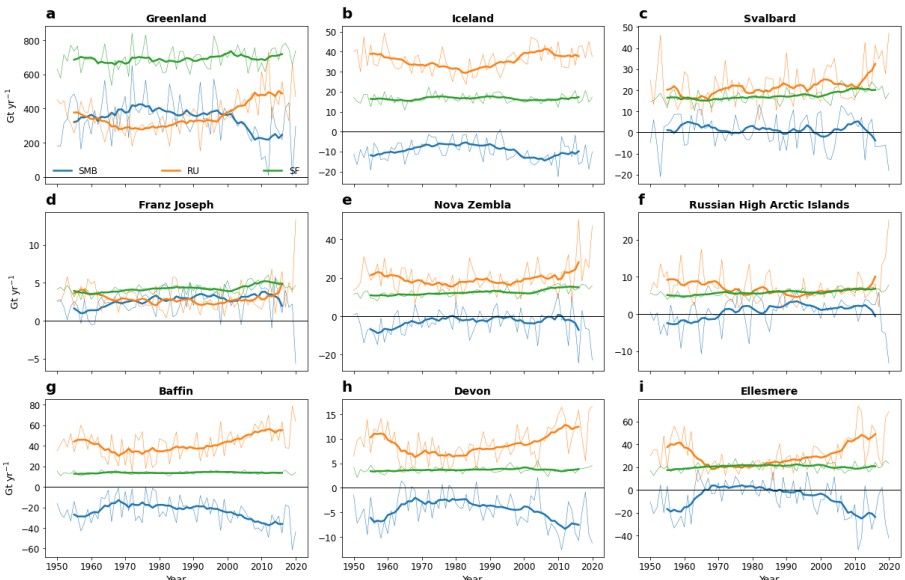

**Figure 4.** Annual (thin line) and 20 years running mean (thick line) of the annual integrated SMB (blue), RU (orange) and SF (green), over (a) Greenland, (b) Iceland, (c) Svalbard, (d) Franz Joseph land, (e) Nova Zembla, (f) Russian High Arctic Islands, (g) Baffin, (h) Devon and (i) Ellesmere.

This difference implies that there is a clear interest in analyzing the different Arctic sub-regions independently to better identify the driving process involved.

## 4.1 Integrated SMB changes

The Baffin Island, Devon and Ellesmere Island ice caps and glaciers have been losing mass since 1950. Over the Baffin Island, it is accelerating in recent years with the SMB going from -22.1 $Gt\,yr^{-1}$ between 1950 and 1970 to -33 $Gt\,yr^{-1}$ from 2000 to 2020. The snowfall has remained stable across the whole period, while the runoff has increased significantly (from 39.6 $Gt\,yr^{-1}$ to 52.8 $Gt\,yr^{-1}$). Further north, the Devon ice cap has seen roughly the same evolution as the Baffin Island. The SMB has decreased from -4 $Gt\,yr^{-1}$ over 1950–1970 to -6.5 $Gt\,yr^{-1}$ over 2000—2020 as a consequence of higher runoff (+2.4 $Gt\,yr^{-1}$) but stable snowfall. The same evolution also occurred over the Ellesmere Island where the 30% increase in runoff leads to a decrease in SMB from -9.3 to -16.8 $Gt\,yr^{-1}$ over 2000–2020 compared to 1950–1970.

Similarly, the SMB has decreased over Greenland, Iceland and Svalbard over 2000–2020 compared to the period before 1970. The strong increase in runoff (anomaly of +119.1 $Gt\,yr^{-1}$) over the Greenland Ice Sheet despite higher snowfall (+42.5 $Gt\,yr^{-1}$) has resulted in a lower SMB (from 343.7 to 267 $Gt\,yr^{-1}$.). Over Iceland, the increase in runoff is not compensated at all by snowfall that remained stable leading to a SMB decrease of 1.4 $Gt\,yr^{-1}$. Over Svalbard, the net SMB was on average positive (1.7 $Gt\,yr^{-1}$) before 1970 but negative (-0.8 $Gt\,yr^{-1}$) after 2000 as a result of an increase in runoff (+8.3 $Gt\,yr^{-1}$).





| Region | Period | SMB | SF | RU |
|---|---|---|---|---|
| | 1950–1970 | -22.1 ±11.8 | 13.6 ±1.7 | 39.6 ±11.2 |
| Baffin | 1975–1995 | -18.9 ±10.6 | 13.9 ±1.2 | 36.8 ±11.2 |
| | 2000–2020 | -33.6 ±10.8 | 13.8 ±1.3 | 52.8 ±10.9 |
| | 1950–1970 | -4.7 ±3.5 | 3.5 ±0.6 | 8.9 ±3.3 |
| Devon | 1975–1995 | -3.1 ±2.1 | 3.7 ±0.4 | 7.3 ±2.2 |
| | 2000–2020 | -6.5 ±3.8 | 3.8 ±0.6 | 11.3 ±3.6 |
| | 1950–1970 | -9.3 ±15.3 | 18.9 ±2.6 | 31.1 ±14.7 |
| Ellesmere | 1975–1995 | 0.4 ±8.5 | 21.6 ±2.8 | 23.5 ±7.3 |
| | 2000–2020 | -16.8 ±16.9 | 20.4 ±3.3 | 41.2 ±16.6 |
| | 1950–1970 | 343.7 ±110.0 | 677.8 ±59.2 | 341.9 ±74.9 |
| Greenland | 1975–1995 | 375.3 ±93.0 | 682.7 ±55.6 | 312.5 ±64.0 |
| | 2000–2020 | 267.9 ±119.6 | 710.3 ±52.5 | 461.1 ±106.2 |
| | 1950–1970 | -2.7 ±3.2 | 12.2 ±1.8 | 25.2 ±4.1 |
| Iceland | 1975–1995 | 0.4 ±4.4 | 13.5 ±1.5 | 21.6 ±4.3 |
| | 2000–2020 | -4.1 ±4.8 | 12.9 ±1.7 | 27.6 ±4.5 |
| | 1950–1970 | 1.7 ±7.6 | 15.8 ±2.5 | 18.9 ±8.7 |
| Svalbard | 1975–1995 | 1.5 ±6.3 | 17 ±2.0 | 20.4 ±7.2 |
| | 2000–2020 | -0.8 ±9.1 | 19.7 ±3.1 | 27.2 ±9.7 |
| | 1950–1970 | 1.9 ±1.2 | 3.8 ±0.5 | 3.3 ±1.3 |
| Franz Joseph | 1975–1995 | 3.1 ±1.4 | 4.3 ±0.4 | 2.5 ±1.4 |
| | 2000–2020 | 2.6 ±2.5 | 4.7 ±0.8 | 3.7 ±2.7 |
| | 1950–1970 | -5.2 ±5.7 | 11 ±1.4 | 19.9 ±6.2 |
| Nova Zemble | 1975–1995 | -0.9 ±5.2 | 12.6 ±1.3 | 17.4 ±5.2 |
| | 2000–2020 | -4.2 ±9.3 | 14.4 ±1.9 | 23.9 ±10.1 |
| | 1950–1970 | -1.8 ±3.5 | 5.1 ±0.8 | 8.5 ±3.6 |
| Russian High Arctic Islands | 1975–1995 | 2 ±3.6 | 5.9 ±0.8 | 5.7 ±3.4 |
| | 2000–2020 | 0.6 ±4.1 | 6.5 ±0.8 | 8 ±4.9 |

**Table 2.** Regional averaged and variability of the SMB, runoff (RU) and snowfall (SF) integrated over permanent ice areas for different time periods, in $Gt\,yr^{-1}$.



|  | r (SMB/SF) | r (SMB/RU) |
|---|---|---|
| Baffin | 0.38 | -0.99 |
| Devon | 0.43 | -0.98 |
| Ellesmere | 0.51 | -0.97 |
| Greenland | 0.63 | -0.85 |
| Iceland | 0.67 | -0.78 |
| Svalbard | 0.21 | -0.89 |
| Franz Joseph | 0.5 | -0.9 |
| Nova Zembla | 0.24 | -0.92 |
| Russian High Arctic Islands | 0.36 | -0.94 |

**Table 3.** Correlation coefficient between annual values of SMB, RU and SF over all sub-regions

On the other side of the Arctic, the SMB has increased over the Franz Joseph Land archipelago, Nova Zembla and the Russian High Arctic Island over 2000–2020 compared to 1950–1970. As a result of higher snowfall (+ 0.9 $Gt\,yr^{-1}$) and a

stable RU, the SMB is now higher over the Franz Joseph Land archipelago. It is also higher over Nova Zembla (-5.2 to -4.2 $Gt\,yr^{-1}$) for the same reasons. Finally, over the Russian High Arctic Island, the SMB has increased steadily from -1.8 to 0.6 $Gt\,yr^{-1}$ because of both an increase in SF (5.1 to 6.5 $Gt\,yr^{-1}$) and a decrease in RU (8.5 to 8 $Gt\,yr^{-1}$). It is the only region where the RU has decreased overall in the simulation period.

The previous paragraphs suggest similar temporal evolution for different SMB components and/or regions. We then present

normalized values of the 20-years running mean SMB, snowfall, and runoff over all regions for a better intercomparison regardless of the size of the different regions (Fig 5.).

Snowfall has increased everywhere from 1950. The Russian Archipelagos (Franz Joseph, Nova Zembla and Russian High Arctic archipelagos) have seen the largest relative increase of snowfall with an acceleration since 1995. To a lesser extent, this can also be observed for Svalbard and Greenland. However, our results suggest that it reached a peak around 1985 over Iceland

and around 1995 for the Canadian regions (Ellesmere, Devon and Baffin islands).

Svalbard excepted, all the regions experienced a large decrease in runoff before a significant increase. Runoff has decreased until 1975 over Greenland and the Canadian Arctic, and until 1985 for the Russian archipelagos. On the contrary, the runoff is steadily increasing throughout the whole period over Svalbard. While Iceland, Greenland, and the Canadian Arctic have experienced an increase in the runoff since 1975, it is clear that the climatological average runoff increase is accelerating

unequivocally in all regions since 2000.

Finally, the SMB evolution can be divided into three main periods over all regions. A first period where it increased as runoff decreased, then a second period with a stabilization (increase in both runoff and snowfall) and then a third where the strong increase in runoff has led to a large decrease in SMB. It is important to mention that Svalbard excepted, all regions had a higher SMB over 1975–1995 than over 1950–1970 or 2000–2020 (Tab. 2).

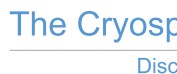
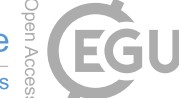


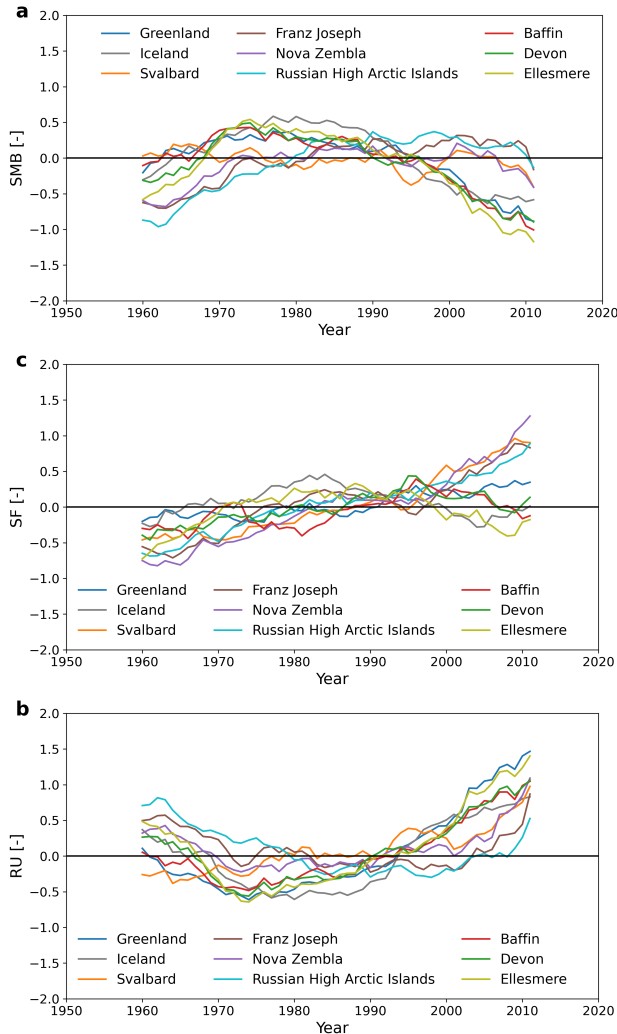

**Figure 5.** 20 years running mean of the normalized timeseries (mean-subtracted and divided by standard deviation) of (a) SMB, (b) RU and (c) SF over all regions.

The SMB evolution is relatively associated with the runoff evolution, only of the opposite sign (see Table 3). This indicates that though snowfall has increased, melt and runoff variations are the main drivers of the recent SMB over the whole Arctic. Greenland, Iceland and the Canadian Arctic are following the same pattern, with a slight increase from 1960 to 1975, followed by a decrease from 1975 to 2000 that accelerates afterward. The Russian archipelagos on the other side have experienced a large increase from 1960 up to 1980, followed by a stabilization between 1980 to 2000, and a slight decrease afterward. Only Svalbard stands out as having a relatively stable SMB (increase in both runoff and snowfall compensating each other) throughout the whole simulation period as already mentioned in previous studies (e.g., Lang et al., 2015).



## 4.2 Spatial tendencies

Generally, glaciers, ice caps and ice sheets tend to see their equilibrium lines (annual SMB equals to zero) rise because of global warming. This tendency is often driven by the increase in surface melt at lower altitudes. This phenomenon can be seen

in Greenland where the ablation zone has experienced a SMB decrease of up to -350 $kg\,m^{-2}\,yr^{-1}$ on average between 1960 and 2000 (Fig 6,a). At the same time, the North East interior of the Greenland Ice Sheet has experienced a SMB increase of +50 $kg\,m^{-2}\,yr^{-1}$ as a result of more snowfall (see Fig.S1).

This tendency is not present over the south Canadian ice caps (Devon, Baffin), where the SMB has decreased nearly everywhere by at least 100 $kg\,m^{-2}\,yr^{-1}$. In Iceland, the Vatnajökull ice cap has seen an overall decrease in SMB, except on its

southern part. Looking at Svalbard, there is a difference between the southern part of the region where SMB has decreased significantly in the ablation zone and the northern, higher, and more icy part of the region where SMB has increased more than 200 $kg\,m^{-2}\,yr^{-1}$ due to larger snowfall (see Fig.S2). Finally, the Russian archipelagos (Franz Joseph, Nova Zembla and Russian High Arctic) have experienced a general increase of SMB over nearly their whole surface, being ablation or accumulation area.

Overall, there is a difference in the SMB evolution between the western part of the Arctic (Canada & Greenland) where the SMB has decreased and the eastern part of the Arctic (Svalbard & Russian Archipelagoes) where the SMB has increased after 2000 compared to the period before 1970. This difference in tendency is very clear between 1950 and 1979, and remains during the recent period, though less pronounced.

## 5 Discussion

### 5.1 Correlation to large scale indices

Between the desertic center of Greenland to the marine Russian archipelagos, the wide variety of climates across the Arctic cryosphere may mean that its response to climate change is not homogeneous spatially. This can be already shown by comparing the climate of the recent past in Greenland (Fettweis et al., 2017) and for example, Iceland (Noël et al., 2022). In the latter region, it has been shown that the North-Atlantic cooling has contributed to stabilizing the SMB of Iceland since 2010, while

over Greenland melt rates were increased by the recurring atmospheric blocking situation gauged by negative NAO conditions.

With the same idea of linking SMB variations to large-scale changes, we selected a wide variety of atmospheric indices, averaged over the whole year, to compare with the annual time series of SMB variables for every Arctic region. Fig 7 shows the correlation of annually-averaged atmospheric indices to summer (JJA) melt (a) and snowfall (b), the main drivers of SMB over the different regions. Two more oceanic indices were also added, the annual average sea surface temperature over 70°N

(SST) and the annual average sea ice concentration (SIC) over 70°N. Overall, a lot of indices do not correlate with the melt rates or snowfall rates.

We see, however, a strong correlation between the melt rates in the Western part of the Arctic (Greenland and Canada) and the GBI and AMO indices. This has already been observed in the recent past (Fettweis et al., 2013, e.g.,) for Greenland. It







**Figure 6.** Annual SMB anomalies between the (1990-2020) and (1950-1979) periods over (a) the whole Arctic, (b) Nova Zembla, (c) Svalbard, (d) Devon, (e) Iceland, (f) Russian High Arctic Islands, (g) Ellesmere, (h) Baffin and (i) Franz Joseph land. Hashed areas denote where the anomaly has a low significance value regarding its variance (using Student's t-test with 90% p-value). The equilibrium line between ablation and accumulation areas for the 1990–2020 period is shown with a dashed purple line.

implies that the blocking situation, which increases melt over Greenland, also strongly impacts the Canadian Arctic. We can
also observe the anticorrelation between the GBI index and snowfall in Iceland. This might be related to the northerly flow induced by the anticyclonic conditions over Greenland when the GBI is high, as it has already been suggested by past studies (Matthews et al., 2015; Fettweis et al., 2013).





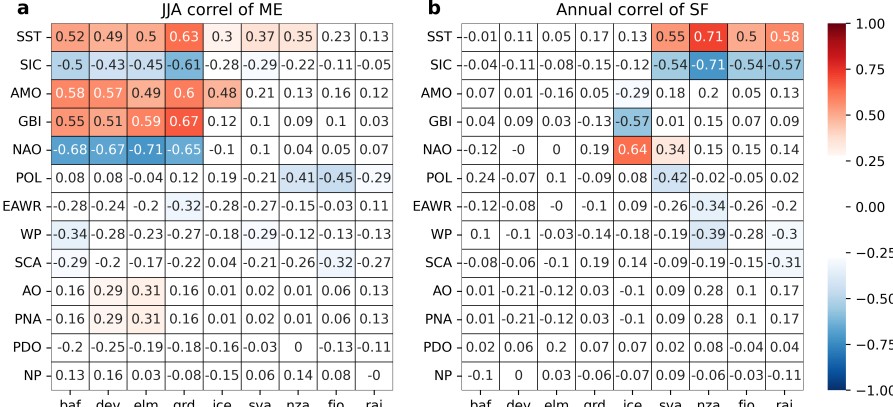

**Figure 7.** Correlation over 1950–2020 of integrated summer (JJA) ME (a) and annual SF (b) over all sub-regions and annual large scale atmospheric/oceanic indices. AMO: Atlantic Multi-decadal Oscillation, GBI: Greenland Blocking Index, NAO: North Atlantic Oscillation, POL: Polar/Eurasian pattern, EAWR: East Atlantic/Western Russia index, SCA: Scandinavian pattern, WP: West Pacific pattern, AO: Arctic oscillation, PNA: Pacific North American index, PDO: Pacific Decadal Oscillation, NP: North Pacific index, SST: Annual average Sea Surface Temperature over 70°N, SIC: Annual average Sea Ice Concentration over 70°N.

On the other side of the Arctic, no such significant correlation to atmospheric indices is found. We see however a common correlation (resp. anticorrelation) between Svalbard & Russian Archipelagos snowfall and average Arctic SST (resp. Arctic SIC). This suggests that a warmer ocean and less ice-covered ocean has likely resulted in higher evaporation and then more snowfall. We found a strong correlation between snowfall and the temperature of the atmosphere around this region (not shown) that also implies higher saturation water vapour pressure and further more precipitation. However, our results do not enable to state if the additional humidity mainly comes from the neighboring ocean or from more remote areas, or a combination of both sources.

Correlating annual values of SMB between all sub-regions (see Fig.8a) confirms the existence of two distinct sub-regions groups of similar evolutions: Greenland along with the Canadian Arctic on one side, and all of the Eastern Arctic from Svalbard to the Russian High Arctic Islands on the other. We see again that the SMB correlation is mainly driven by ME. We also see that SF correlate more between the regions of the Eastern Arctic then the Western Arctic, noticeably between Nova Zembla, Franz Joseph Land and the Russian Arctic High Islands.

## 5.2 Comparison with mass balance & calving estimates

As we studied processes taking place at the surface of permanent ice areas, integrating the SMB spatially does not reflect the total ice Mass Balance (MB). More specifically, the increase in SMB does not imply an increase in ice mass as altimetry and gravimetry measurements demonstrated that those regions still lose mass. Though the scale is much lower than in the Antarctic,





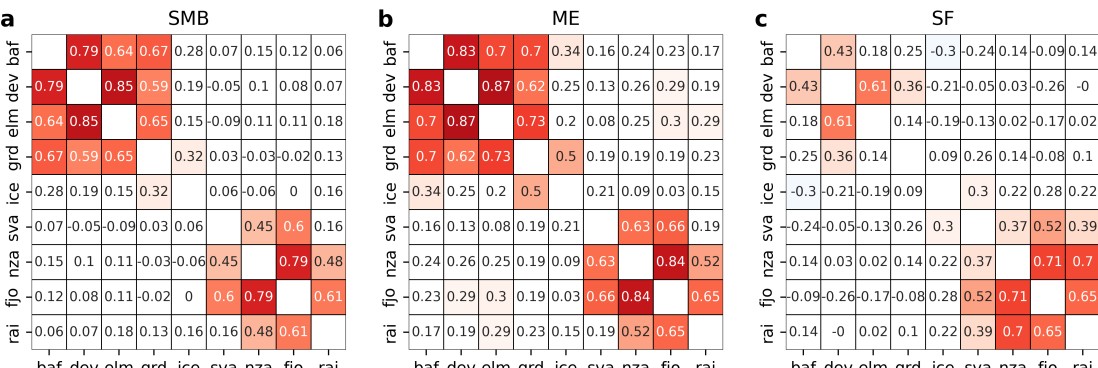

**Figure 8.** Inter-regional correlation of 1950–2020 annual SMB (a), ME (b) and SF (c)

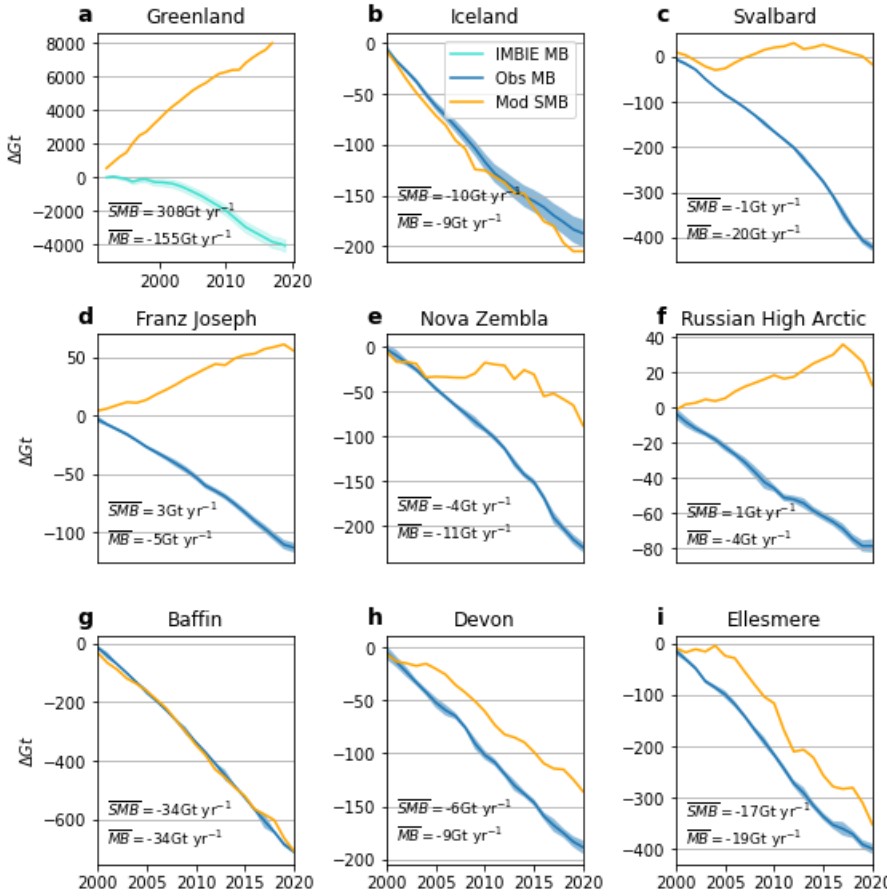

**Figure 9.** Cumulative modeled SMB and observed MB from 1990 to 2020 for (a) Greenland (using IMBIE dataset) and from 2000 to 2020 for (b) Iceland, (c) Svalbard, (d) Franz Joseph land, (e) Nova Zembla, (f) Russian High Arctic islands, (g) Baffin, (h) Devon and (i) Ellesmere.





ice calving can make up large proportions of the total ice loss (sometimes called dynamic ice loss) in some Arctic subregions.

For example, ice discharge was roughly equal to melting in Greenland between 2008 and 2012 (Enderlin et al., 2014). To assess the importance of calving against SMB over the different regions, we compare our SMB estimates to integrated MB products. Over the Greenland Ice Sheet, we used the IMBIE Mass Balance dataset (Team, 2020). It consists of mass change measurements from satellite gravimetry and satellite altimetry from 1997 up to 2012. For the other regions, we used the altimetry dataset of (Hugonnet et al., 2021) mentioned previously.

By comparing the cumulative MB and the cumulative SMB (Fig.9), we can estimate the calved volume over all the subregions of the study. Over the Canadian Arctic (Devon, Ellesmere and Baffin), the dynamic ice loss is relatively low (even close to zero in the case of Baffin), thus the ice mass loss can be considered as mainly driven by the surface mass balance and then the atmospherics conditions. This low dynamical ice loss can be explained because only a few glaciers are marine terminating in Baffin Island. It is however not the case over Ellesmere and Devon Islands, where the surface ratio of marine terminating

glaciers is close to 50%. There, the low dynamical ice loss could be explained by the SST of the waters surrounding the North Arctic Canada that has not significantly warmed yet, compared to atmosphere temperature over the glaciers. Contrarily in the eastern Arctic, while the SMB continues to be positive over Franz Joseph or Russian High Arctic Islands and has overall increased since 1950 (see Fig.6), the ice mass is still decreasing rapidly (up to a MB of -5 $Gt\,yr^{-1}$ over Franz Joseph land). It is also the case over Svalbard and Nova Zembla, where a relatively constant SMB since the beginning of the 21st century goes

along with a steady decrease of the total ice mass. This can be explained by the rapid Arctic Ocean warming that increases the calving rates rapidly, particularly near Svalbard and Nova Zembla, where its warming is the most pronounced with more than 0.8°C per decade (Li et al., 2022). Note that while the Greenland Ice Sheet SMB is positive, lower recent values have resulted in stronger mass loss as highlighted by Fig. 9,a. Greenland aside, the average Arctic ice MB has been of -111 $Gt\,yr^{-1}$ since 2020, while the average SMB has been of -62 $Gt\,yr^{-1}$.

## 6 Conclusions

Considering all the land ice over the Arctic, our simulations reveal that the annual surface mass balance has decreased by 120 $Gt\,yr^{-1}$ between the period of 1950-1979 and 2000-2020. This overall mass loss has been accelerating by -4 $Gt\,yr^{-2}$ from 2000. It is mainly driven by melt, which has on average increased by 21% since 1950. This melt increase is however heterogeneous spatially, with an increase of 41% for Greenland, but only 9% on average over the Russian sub-regions where

snowfall accumulation has increased by 28%. Along with Svalbard, those regions have experienced a general increase of their SMB when looking over the whole simulation period. However, record low SMB have been observed everywhere during the last decade, such as in 2020 for all of the Eastern Arctic, Devon and Ellesmere, or 2019 for Greenland and Devon.

We have also identified two distinctive sub-regions groups (Baffin, Devon, Ellesmere, Greenland; and Svalbard, Franz Joseph, Nova Zembla & Russian High Arctic) that seem to have the same links to climatological drivers and that went un-

der a comparable SMB evolution. We have shown that melt is correlated to GBI over Greenland and North Canada. Snowfall over the latter group seems to be correlated to the average Arctic ocean temperatures, while it is not the case elsewhere. No

atmospheric large-scale indices seem correlated to its evolution. While these links have been established for the annual mean SF and ME time series, more work remains to be done to understand what is driving the surface mass balance over those two groups. This is especially the case over the Russian Arctic, where only a few studies have been carried out.

Finally, our results suggest rapid changes in the Arctic land ice. While some regions in the Arctic have gained mass at their surface (but still losing mass taking into account the ice dynamics), these conclusions could be totally different in the years to come. For instance, most recent years were marked by several negative records over the Russian sub-regions. A repeat year of such extreme melting could quickly reverse the trend in these regions and lead to a general loss of surface mass throughout the Arctic. It will therefore be important to continue to study the Arctic land ice and to update these results regularly.

*Code and data availability.* Observational data were downloaded on different institutes and organizations web sites. For the Canadian Arctic, the AWS data was provided by the Government of Canada (https://climate.weather.gc.ca/historical_data/search_historic_data_e.html); by the Norwegian Meteorological Institute (https://seklima.met.no/stations/) over Svalbard, by the the Russian Meteorological Institute for the Russian Arctic (available at https://www.ncei.noaa.gov/access/search/data-search/global-hourly). Over Iceland and Greenland, we used the compiled datasets of European Climate Assessment & Dataset (ECAD, https://www.ecad.eu/dailydata/customquery.php).

The MAR code used in this study is tagged as v3.11.5 on https://gitlab.com/Mar-Group/MARv3.7 (MAR Team, 2021). Instructions to download the MAR code are provided on https://www.mar.cnrs.fr (MAR model, 2021). The MAR version used for the present work is tagged as v3.11.5

*Author contributions.* DM, CK and XF designed the study. XF ran the simulations. DM made the plots (benefits from some scripts of CK), performed the analysis and wrote the manuscript. CL provided help with gathering and processing the observational data. CK, AD and XF
provided important guidance while all the authors discussed and revised the manuscript.

*Competing interests.* X.Fettweis is an editor of The Cryosphere.

*Acknowledgements.* We acknowledge the PolarRES european H2020 project (program call H2020-LC-CLA-2018-2019-2020 under grant agreement no. 101003590) for funding this paper. Computational resources have been provided by the Consortium des Équipements de Calcul Intensif (CÉCI), funded by the Fonds de la Recherche Scientifique de Belgique (F.R.S. – FNRS) under grant no. 2.5020.11, and the
Tier-1 supercomputer (Zenobe) of the Fédération Wallonie Bruxelles infrastructure, funded by the Walloon Region under grant agreement no. 1117545.



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
