# Peer review of "Spatially heterogeneous effect of climate warming on the Arctic land ice"

_The Cryosphere, 2023_

## Author Comment (AC1)

Dear Shawn Marshall,

Thank you very much for your review and suggestions that will certainly improve the quality of our manuscript. As you suggested, the question of decoupling Polar amplification vs synoptic circulation is indeed very interesting, though it may be worth another paper.

We have written answers to your comments in blue.

Best,

Damien Maure and all authors.

l.18, "could modify the SMB" - I guess this is very clearly happening, beyond just conditional. Many prior studies show how changing T and P are modifying the SMB across Arctic ice caps, e.g. Huggonet et al. (2021), IPCC (2021) and references therein

Absolutely, thank you for the notice there's too much caution there. We will correct this in the revised version of our manuscript.

l.23, "quick" is hard to define - seconds, minutes, months, years. Suggest being specific here, e.g., if you are referring to synoptic, seasonal, or interannual variability

It might be more appropriate to write "seasonal climate variations" indeed.

l.28, see also Rajewicz and Marshall (2014) on this point. Not that this needs to be cited, but it directly assesses the anticyclonic circulation/ridging anomalies that are being discussed here, and notes how these strongly and simultaneously impact Arctic Canada and southern/western Greenland, of relevance to this manuscript. I would also note that this can be expected to be highly correlated with cool anonalies in the eastern subArctic, as ridging over the western Arctic and Greenland would typically be accompanied by a trough (cooler conditions) in the eastern North Atlantic and Eurasian sector of the Arctic

Thank you for your suggestion, the paper is interesting and we will add it to the list of reference supporting the evidence of this strong effect of anticyclonic blocking over Greenland. It is also interesting to integrate it in the discussion, as we find a strong (anti)correlation between GBI and NAO, suggesting your point (in Rajewicz and Marshall 2014) about ridging patterns and NAO being covariant.

l.53, the 6 km resolution is high in some ways, for the size of the domain, but does not resolve many of the smaller ice masses, particularly in mountainous regions such as coastal Greenland and Baffin Island. On this particular point on l.53, omitting grid cells that are less than 50% ice covered, I worry if

this might exclude a large amount of the ablation area of many of the glaciers and ice caps. This could cause a systematic underestimation of ablation, by excluding a lot of marginal ice area. It will be good to discuss this and even compare the captured ice area/hypsometry to what one would see at 1 km, for example.

This was a comment made by the editor, absolutely relevant. We have done exactly what you suggested, comparing the hypsometries of the ice area (Fig. S1, see below). We find no significant discrepancies, except a small lower elevation for Franz Joseph Land and Baffin Island. This figure will be added in the supplementary of our revised manuscript.

[Figure]

Figure 2, "annual time scale" - are these averaged for the decade? Please clarify in the caption

Thank you for the notice, this was unclear. We will improve this sentence by "Distribution of AWSs 10-years correlation coefficient between observed and modeled values." in the revised manuscript.

l.137-138, discussion of the lower interannual variability of the altimetry data. This would be helpful and interesting to compare with WGMS SMB data which is available for some of these regions (e.g., Artctic Canada, Iceland) - what does the interannual variability look like there? It would be very instructive to include a third box-whisker for the WGMS data where it is available, recognizing that it is not covering the full domain in any of these regions. Particularly around whether the modelled interannual variabiliy is realistic, and to compare SMB with SMB directly for all regions where this is possible.

This is a good idea, and we have done an evaluation using WGMS prior to switching to altimetry because of a low coverage over certain regions. Nevertheless, we have formatted the results as you suggested:

[Figure]

The figure shows the annual mean specific mass balance over different glaciers of a given region, with *n* the number of observations (Greenland periphery and Franz Joseph land are not included because there is no observation).The mean annual values and interannual variabilities are relatively close. The variability of the observations are closer to modelled values than in the case of Figure 3 of the manuscript with the altimetry dataset, in line with our comment on the lowered variability of such satellite products.

Overall, we need to keep in mind that only a small fraction of the iced area is evaluated here (except Iceland where a significant portion of the iced area in included). This could explain the strong underestimation of the SMB vs observations available over Arctic Canada (noticeably over Baffin islands), though in line with your comment on l.157.
You will find below the detail of all observations available of the WGMS, compared to our MAR outputs, with the orange line being the modelled SMB and the blue line being the observed WGMS SMB, in mWE.yr[-1]. This figure will be added in the supplementary of our revised manuscript.

[Figure]

l.151-152, "while this decrease is mainly driven by Greenland..." True, but this is mostly because Greenland dominates the total mass loss? vs. the % change being the driver, as argued here.

We are not sure of what you mean here, but we also realize our sentence might not be clear. IWesuggest rephrasing by "This total SMB decrease is mainly driven by Greenland (as being by far the largest ice body). However, Greenland runoff has increased by 35% between (1975–1995) and (2000–2020), but has on average increased by 45% over the other regions."

Figure 4, Please define RU and SF in the caption

Thank you for the notice it should indeed be defined.

l.157, Are these numbers right, for Baffin Island? Something is sending up red flags for me here. The glacierized area of Baffin Island is much less than Ellesmere, so the modelled runoff and mass loss from here seems out of proportion compared with Devon and Ellesmere. There are a lot of smaller ice masses that may not be well-captured at 6 km. This might make sense in the context of more negative specific mass balance rates here (average m/yr of thinning), but it would be helpful to discuss and present this for the different regions, based on the RGI glacier areas.

Good point. It is true that, compared to the size of the glacierized area, the numbers are big. However, as seen in Fig.6, we model a decrease in SMB over time stronger above Baffin Island than over Devon or Ellesmere (with close to zero accumulation area). Moreover, it is comparable to what Noël et al. (2018) found (close to -30Gt.yr-1 if you look at the 2000-2020 period in their Fig.5(b), but also strong melting since the 60's.), and we suggest adding a reference to that paper in the sentence l.157. Recent recurring blocking events over Greenland tend to increase the melt even more with strong positive temperature anomalies over Baffin Island.

It is still might be worth adding a recent value of m/yr per region as you suggest, though it is already written in Fig.3 (land terminating only) for the recent (2000-2020) period.

l.248, "that those regions" - Do you mean the eastern Arctic? Be specific here.

Again thank you for the notice, this is a bad phrasing. This will be "that all the regions studied here are still loosing mass" in the revised version of our manuscript.

l.264, I think that here and throughout, this should be Novaya Zemlya. Nova Zembla is an island in the Canadian Arctic, near Baffin Island, but is not what the authors are referring to, I think

We did not know the existence of this other island. We guess Novaya Zemlya is sometimes also called Nova Zembla, but indeed it is clearly worth changing for clarity in the revised version of our manuscript.

l.34, suggest rewording, "a unified estimate is still lacking"

l.39, "aims", plural

l.66, "over the ocean"

l.67, suggested rewording to "surface pressure, sea ice concentration, and sea surface temperature"

l.156, "over Baffin Island" (no the, here and throughout)

l.216, I don't think "desertic" is a word. Recommend just "dry" ?

l.268, Fig. 9a

l.269, "has been -62 Gt/yr"

Thank you for all the rewordings and corrections above. We will include them in the manuscript.

---

## Author Comment (AC2)

Dear Referee,

Thank you for your comments and suggestions improving our study. We understand your concern about stake observations not being used. We have contacted the owners of the Svalbard stakes dataset to compare to our model results. You will see below our response and evaluation made using the suggested stake dataset. We have also, answering comments to Shawn Marshall (RC1) added a glacier-specific SMB evaluation using WGMS dataset, though it does not concern Svalbard.

You will find answers and/or additions to the manuscripts below each of your comments, in blue.

Best

Damien Maure on behalf of all authors.

L2-3: Svalbard experienced record melt in summer 2022.

To make it more precise, we will change this sentence to "The last two decades were notably marked by melt records over the Greenland Ice Sheet while other regions such as Svalbard seem to remain less influenced." in the revised version of our manuscript

L11: The 9% increase in melt is much lower than other studies have shown for Svalbard (e.g. Östby et al. 2017, Van Pelt et al. 2019, Noël et al. 2021). This should at least have been acknowledged since other studies focusing on individual regions did more efforts to calibrate their models against available data (primarily stake data).

This sentence of the abstract is indeed misleading. If you look in the study, we found in Svalbard ~18.9Gt.yr-1 runoff rate for the 1950-1970 period, that has increased to ~27Gt.yr-1 in 2000-2020 (making it a 43% increase). The 9% comes when including the other Russian Islands where we find that melt has indeed not been increasing a lot. We agree that this way of writing is not relevant and making like Svalbard does not melt was not our goal. We will replace this sentence by "This increase is not seen over the Russian Arctic permanent ice areas, where the total melt rate has increased by only 3%".

L20-21: To be complete also mass fluxes by condensation/riming should be considered in the SMB.

Condensation and rimming are integrated in our model in the sublimation/evaporation variable. But we will add this to the sentence to precise that point.

L31-32: Changes in SMB in Svalbard were maybe not as large as e.g. in the Canadian Arctic, but they were still significant. See Schuler et al. (2020; doi: 10.3389/feart.2020.00156) and references therein. Except Lang et al. 2015 all other modelling studies found significant negative mass balance trends.

We understand your concern about our phrasing minimizing the changes in Svalbard. It is meant to be compared with what we can see in, as you said, the Canadian Arctic for example. We suggest rephrasing this as "atmospheric circulation changes has tempered significant changes in SMB" in our revised manuscript.

L60-63: A 6-km spatial resolution is a reasonable choice given the large simulation domain. It could be acknowledged here though that a lack of topographic detail affects calculation of surface mass balance in areas with strong topographic variations, particularly through impacts on local precipitation, wind drift, temperature, insolation and shading.

Thank you for your comment. We suggest adding a sentence l.63 "Even though the hyspometries do agree, the model resolution can affect the surface mass balance in strong topographic variations areas, affecting shading, wind drift and precipitation."

L73-76: It is unfortunate that no stake data were used in this study for a region like Svalbard, whereas they are readily available. Comparing only against geodetic mass balance for land-terminating glaciers gives a somewhat biased assessment (since large tidewater glaciers and ice caps are excluded). And more importantly, the geodetic data do not allow for validation of temporal variability and trends of SMB, which would have been possible with stake data.

The figure below presents our SMB evaluation over Svalbard using the stakes dataset that has been used by Noël et al. (2020). To make an appropriate comparison not biased by an altitude difference between the MAR grid and the stakes, the modelled SMB was downscaled to the stakes altitudes using a local SMB altitude gradient, with the methodology described in Franco et al. (2012) or Fettweis et al. (2020).

[Figure]

Here we see that there is a good agreement between modelled and observed values. The observed stake-averaged interannual-variability is 0.32 mWEyr-1, while the modelled one is of 0.38mWEyr-1, and the average SMB correlation on a given stake is 0.69.

Table 1: It would have been great to see a comparison of linear trends (particularly in T2m) as well.

Please see answer to comment on L156-162.

L109-111: Why use data corrected to sea level when raw data exist too?

See our answer below: this is a mistake and it will be changed in the manuscript, thank you for pointing this out.

L112-113: Is the negative bias of >3 degrees C for Svalbard also a result of a sea level correction? If not, such a temperature could have a pronounced impact on melt in summer.

There is indeed an error in this sentence. Thank you for pointing this out, this was not a problem of corrected data but of altitude bias: over Svalbard, available AWS are located very close to the sea, so they have a very low elevation, while our model topography with 6km resolution are generally above. (The same is true for peripherial AWS in Greenland). See below the altitude bias, and a first order correction of the temperature bias using a dry temperature gradient of -1°C/100m in summer:

| Station | Station altitude [m asl] | MAR altitude [m asl] | Summer bias [°C] | Altitude corrected summer bias [°C] |
|---|---|---|---|---|
| Hopen | 6 | 11 | -0.63 | -0.58 |
| Sveagruva | 9 | 244 | -4.62 | -2.27 |
| Lufthavn | 2 | 298 | -5.06 | -2.1 |
| Ny-ålesund | 8 | 180 | -2.53 | -0.81 |

We see here that by compensating this altitude bias, the temperature underestimation is reduced, whist still being lower than the observed temperature. However, the comparison with stake data above suggests a good comparison to observed values. We could assume that this cold bias along the sea does not impact a lot on the melt and SMB inland. Finally, it should be noted that the oceanic conditions (SST and sea ice cover) in MAR comes from ERA5 and is likely not representative of the sea temperature along the coast impacting on measured temperatures.

Figure 2: Pressure is maybe not the most crucial parameter to validate here, as it has hardly any impact on mass balance calculations. On the other hand, precipitation is important but not validated at all. For example, stake winter balance data could have given an indication on potential biases of snowfall during the cold season.

We understand your concern about pressure, but it is one of the most important variable to validate when dealing with regional atmosphere modeling. This is to ensure that the model is reproducing the real atmospheric circulation, which is the first step to produce accurate precipitation and melting rates. On the other hand, it is very difficult to obtain continuous data on precipitation at the scale of the whole arctic (even stakes only give the annually integrated SMB).

Section 3.2: I appreciate the use of geodetic MB observations for validation. There are however some drawbacks too. Besides that only land-terminating glaciers can be compared it also does not enable comparison of temporal mass balance variability and trends. This would have been possible when (also) stake data would have been used for comparison. It is particularly important in this study which draws conclusions on differences in mass balance and trends between regions.

In addition to MB observations, we have also used SMB observations (the ones used in Fettweis et al., 2020) over Greenland. But, to make the evaluation more robust as you suggest, we will also add

an evaluation using the stakes of Svalbard, and we will also add an evaluation using WGMS data where it is available.

MB satellite estimates have also the benefit of looking at glacier-wide SMB, which is easier to compare to 6km pixels in our models. Point-stake data can suggest local SMB variations not resolved in our model (as you pointed out).

Figure 3: Two things are interesting here: 1) the larger spread in model results per region than the geodetic observations show, 2) the stronger region to region variability in the model than in the observations. This would be an interesting discussion point, that I think should be added in the Discussion session.

We briefly address the point of the larger spread of the model L137. For the region-to-region variability, we agree that the model spread looks larger, but we feel it is more a consequence of the first point.

L156-162: It would be more robust to calculate a (linear) trend based on all results since 1950. Now the significance drops because of the use of a selection of data. The results may also be biased a bit by excluding a relatively cold period 1970-1990 in many Arctic regions.

On the point of the linear trend, we deliberately chose not to do so because while some evolution is relatively linear over e.g. Svalbard, it is not the case in the other regions. (because mainly of, as you say, the colder period of 1970-1990). To support our point, there is below a table of the $R^2$ values for a linear trend computed over the whole period: as you can see, those ones are very low.

| Region | Linear fit $R^2$ |
|---|---|
| baf | 0.071 |
| elm | 0.036 |
| dev | 0.034 |
| grd | 0.083 |
| ice | 0.007 |
| sva | 0.013 |
| nza | 0.004 |
| fjo | 0.014 |
| rai | 0.043 |

Understandably, what would be maybe interesting is to compute linear trends over the 3 periods we used to discuss the results (1950-1970; 1975-1995; 2000-2020).

On the point of "biasing" the results, there is a choice to put some numbers in light and not others in the text, but we feel it is always clear what period we are comparing in every sentence (full results can be seen in Table 2). Furthermore, the full time series are available in Figure 4 and provide full information of the SMB evolutions.

L168-173: Here it would have been good to include comparisons with other (region-specific) studies (e.g. Schuler et al. 2020 for Svalbard). This would help because the validation against the geodetic data does not give any insight in reliability of simulated surface mass balance trends.

Thank you for pointing this out. Here is a comparison with the main studies integrating mass balances we have found over the different regions we looked at. It is however important to note that

the considered ice sheet area is different in the different studies impacting the comparison.

| Study | Region | Period | Average SMB [Gt.yr-1] | This study [Gt.yr-1] |
|---|---|---|---|---|
| Noël et al. 2018 | Canada | 1958-1995 | -20.2 | -24.4 |
| Noël et al. 2018 | Canada | 1996-2015 | -46.6 | -49.8 |
| Fettweis et al. 2020 | Greenland | 1980-2012 | 338 | 325 |
| Lenaerts et al. 2013 | Canada | 2004-2013 | -64 | -60 |
| Noel et al. 2022 | Iceland | 1958-1994 | -1.4 | -4.7 |
| Noel et al. 2022 | Iceland | 1995-2010 | -10.3 | -13.7 |
| Noel et al. 2020 | Svalbard | 2013-2018 | -19.4 | -3.9 |
| Noel et al. 2020 | Svalbard | 1958-1985 | 6.3 | 2.25 |
| Radic and Hock, 2011 | Svalbard | 1961-2000 | -1.36 | 1.8 |
| Van Pelt et al.2019 | Svalbard | 1957-2018 | 3 | 1.4 |
| Aas et al. 2016 | Svalbard | 2003-2013 | -8.7 | 2.3 |
| Lang et al, 2015 | Svalbard | 1979-2013 | -1.6 | 1.3 |

Our study remains close to the literature overall. The only exception is Svalbard where as you already pointed out there are strong differences between studies, mainly during the XXIth century period. (We see for example, for the ~1960-2015 period, a positive SMB for Van Pelt et al. (2019) but a negative one in Radic and Hock (2011)). This point will be acknowledged in discussion and conclusion of our revised manuscript.

(There is, as you can see, nothing we could find over the Russian Arctic.)

L195-196: Lang et al. (2015), which also uses the MAR model, is the only study out of many in recent years that also simulated stable mass balance. This discrepancy with other literature should be acknowledged.

This sentence is to be contrasted with the rest of the study. Over the recent period, we see a decrease in SMB, but over the whole simulation period, the decease remains way lower than the inter-annual variability, which is not the case over all other regions (that is why we wrote "relatively stable"). Though, as the trend over Svalbard is one of the only to be linear, we will add this sentence "While it is more stable than other regions, we still find a SMB linear trend of -0.04 Gt yr$^{-2}$ over the whole simulation period".

As shown in the table of our comment above, we find a clear decrease of SMB comparing 2003-2013 to 2013-2018 in our study. The positive bias could be linked to the domain resolution affecting the altitude of the considered iced area, in relationship with the negative temperature bias.

L210-212: See also my earlier comment. It is somewhat biased to compare 1950-1970 to 2000-2020 and not 1975-1995 to 2000-2020, or 1950-1985 to 1985-2020. By the way, why the 5-year gaps? Please note that the increase of SMB after 2000 for Svalbard does not agree with the consensus results by Schuler et al. (2020).

Thank you for pointing this out. In L211, it should only be mentioned "(Russian Archipelagoes)" and not Svalbard. We found a decrease in SMB that relates to what Noel. et al. (2020) found (-0.8 Gt/yr after 2000).

On the point of biasing, as mentioned earlier, the averages are available for every period in Table 2.

The 5 year gap is here to ensure we have periods of the same duration. (This 3-period of analysis was chosen because of the general SMB/RU evolution over a lot of regions, with a relatively colder period around 1980 that made a trend analysis less appropriate). It can be seen as arbitrary but it is only an indication, as full time series are available in Figure 4. If the editor requests it, we can add a table with 20 years moving averages values for every region.

L250-254: The frontal ablation dataset for Northern Hemisphere tidewater glaciers by Kochtitzky et al (2022; doi: 10.1038/s41467-022-33231-x) could have been of use here.

Thank you for your suggestion, we did not know about this dataset. We have looked on it but for the moment we are not sure whether it is worth including as the goal of the study is not to produce a calving estimate. So we will rephrase the subtitle stating: "Mass balance comparison & calving rates." in the revised manuscript.

Title: Please consider removing "the" before "climate warming"

L18-19: Remove brackets around "i.e. the Greenland ... perificial glaciers".

L22-23: Remove brackets around "Note that ... Cogley et al. (2010)"

Table 2 caption: "averaged" --> "averages"

L206: "more icy" --> "colder"

L248: "lower" --> "smaller"

Thank you for those corrections. That will be included in the revised manuscript.